# R&D Expenditure for New Technology in Livestock Farming: Impact on GHG Reduction in Developing Countries

**Alessia Spada ***[ID]**, Mariantonietta Fiore, Umberto Monarca and Nicola Faccilongo**

Department of Economics, University of Foggia, 71121 Foggia, Italy; mariantonietta.fiore@unifg.it (M.F.);
umberto.monarca@unifg.it (U.M.); nicola.faccilongo@unifg.it (N.F.)
**\*** Correspondence: alessia.spada@unifg.it

**Abstract:** The achievement of the objectives of reducing greenhouse gas (GHG) emissions has increasingly received attention and support from decision makers and research by scholars. The livestock sector has always been one of the major sources of GHG emissions, especially in developing countries that do not have green technologies to improve the management of livestock waste. In order to achieve an absolute reduction in emissions, developed countries have applied a wide range of mitigation options; however, there are few studies from the developing world, although greenhouse gas emissions in developing countries have registered a rapid growth. Therefore, this research aims to assess and understand whether public R&D investments can affect emissions deriving from the livestock sector in developing countries. We made use of the FAOSTAT data (FAO Statistical Databases United Nations) and ASTI data set (Agricultural Science and Technology Indicators), collecting data from 29 Africa countries, in 2014 (latest data available). The data were analyzed by means of a Generalized Propensity Scores (GPS) approach, an increasingly widespread technique that is more robust than regression models, especially in small datasets. Our analysis suggests that the livestock sector in these countries shows an improvement in its relationships with the environment and GHG emission levels when the level of public R&D (Research and Development) investment on agriculture is greater. Therefore, reducing greenhouse gas emissions by investing in research and development can lead to more efficient and sustainable resource management for developing countries.

**Keywords:** sustainability; R&D government policy; GHG emissions; Generalized Propensity Score; developing countries; Africa

## 1. Introduction

Agricultural and forestry sectors are characterized by direct $CO_2$ emissions and methane and nitrous oxide emissions. In the specific case of breeding, the majority of direct emissions are methane emissions. In particular, the worldwide dairy sector backs 4.0% of the global anthropogenic greenhouse gas (GHG) emissions [1] and cattle farming for dairy is one of the main important agricultural activities in the world.

In general, GHG emissions in developing countries are increasing more and more, whereas such emissions in developed-countries are decreasing [2]. In addition, emissions and climate changes are strongly linked, and the former could have important implications on agriculture and the well-being of the population in developed and developing countries and even more severe consequences for the latter [2].

Climate change is already acting on all dimensions of food security, availability of food and stability of its supply, its access and its use, both on a local and global level. The most vulnerable groups

are the poor and small farmers. Therefore, policy makers should address research and finance new technologies aimed at reducing emissions more than intervening in emergencies, as these also have a high cost on human lives. Climate change has hit Africa countries with extreme hardness, undermining its vulnerable agricultural sector, which accounts for 70% of the population. The phenomenon also have significant consequences on the development and economic growth of Africa, since climate change uncertainty represents a barrier to investment, complicating long-term planning and infrastructure design [3].

In this paper, we investigate developing countries in Africa with the aim of evaluating if R&D expenditures (through ASTI, Agricultural Science and Technology Indicators) can be effective in reducing GHG emissions of cattle dairy. This research question is crucial for research but above all for policy makers, because a more advanced and sustainable agriculture on the one hand reduces the country's food deficits and, on the other hand, it allows the country to have a high environmental performance. Therefore, we analyze the impact of R&D expenditures of the agricultural sector on GHG emission of cattle dairy in 29 developing countries. The analysis of the evaluation of an impact is increasingly important, especially when it is necessary to evaluate the impact of public investments, often characterized by scarcity of economic resources, as in the case of developing countries. Several methods have been developed to assess the impacts of economic policies. The Generalized Propensity Scores (GPS) approach is an increasingly widespread technique as it is more robust than regression models especially in small datasets. The GPS produces an estimate of a dose-response function. Therefore, the paper estimates the impact of R&D investments (variable treatment) on the GHG emission of cattle dairy (variable outcome) expressing it in terms of dose response function.

The present work is structured as follows. The next section analyses literature and policies on the topic; then, the work describes the data and GPS procedure. Results are presented and discussed. In conclusion, policy implications and potential extensions of research are presented.

## 2. Literature Review

Research focusing on the correlation between economic growth and ecological degradation analyzes the Environmental Kuznets Curve (EKC) [4]. EKC hypothesizes that when a country starts its industrialization process, as its national income increases, so do the levels of pollution, due to the more intensive use of natural resources. As a result of further increases in national income, there is a growing awareness among citizens of environmental quality and, as a result, a growing consumer's willingness to pay for products with a lower environmental impact. Therefore, at this stage, an additional rise in income level determines a decrease in pollution. Grossman and Krueger [5] support the EKC hypothesis, but subsequent empirical studies have mixed results, in particular on the income level where environmental degradation should go towards decline [6–11]. On the same subject, Arrow et al. [12] state that economic growth is sustainable in the long-term only if it is compatible with environmental quality, but even in this case the literature has not been able to produce a unique result. Some studies support the relationship between economic growth and deteriorating environmental quality [13–16], while other scholars show that the link between income growth and environmental degradation is insignificant [17,18].

According to Ansuategi and Escapa [19], the failure of ECK in the field of GHG emissions is due to their particular nature: they create a global and not local disutility, so consumers and policy makers underestimate the consequences of increasing GHG emissions and instead focus their attention on economic growth.

In order to resolve this mismatch, some scholars have focused their attention on specific causes of increased GHG emissions. Part of the literature has focused on the production of electricity as the main cause of GHG emissions, thereby exploring the relationship between economic growth, growth in electricity consumption and GHG emissions. Starting from the pioneering work of Kraft and Kraft [20], some scholars have investigated the relationship between energy consumption and economic growth, with complementary empirical results among studies that highlight an unidirectional

relationship between these two variables [21–23] and studies that instead indicate a bi-directional randomness [24–28]. Finally, some scholars also argue that there is no correlation between electricity consumption and economic growth [29,30].

In addition to the study of the general relationship between economic growth and environmental sustainability, the literature has also deepened the territorial aspects of this relationship [31–34], specifically several studies have focused their analysis on the BRICS (Brazil, Russia, India, China, and South Africa) [35–37].

However, there are few studies that address this concern in developing countries [38]. This is mainly due to the fact that literature is focused on the evolution of green technologies in industry and manufacturing [39], while in developing countries agriculture remains the sector that contributes most significantly to the domestic product and is also the most representative sector of the green economy [40].

Indeed, agriculture is the main and crucial sector of all African countries, as it can try to eradicate poverty, undernourishment and inequality, promote investments and food security, strengthen diversification and address green development processes. The Comprehensive African Agricultural Development Program (CAADP) belongs to the New Partnership for Africa's Development (NEPAD) because agriculture represents a prominent sector of the total GDP [41]. The average value added in the agricultural sector of Africa as percent of GDP in 2018 was 18.23% and over 50% of the total labor force is in this sector (Figure 1). If we consider the value added in the agricultural sector in the rest of the world, it is possible to notice a huge difference between the latter and i.e., the Sub-Saharan Africa countries (Figure 2). In particular, livestock is already 51% of the agricultural economy in South Africa where live cattle are the main traded products. The World Bank [42] foresees that agriculture in Africa will increase to become a US$1 trillion industry by 2030 and correlated with this growth is be a strong increase in GHG emissions. In Africa countries, population growth, with the consequent growing demand for food, also creates a strong pressure on GHG emissions [43] due to the intensive use of fertilizers [44]. Therefore, one of the main challenges for Africa will be to increase production and productivity by implementing tools for the sustainable management of resources and correlatively by controlling impacts on the environment and climate change.

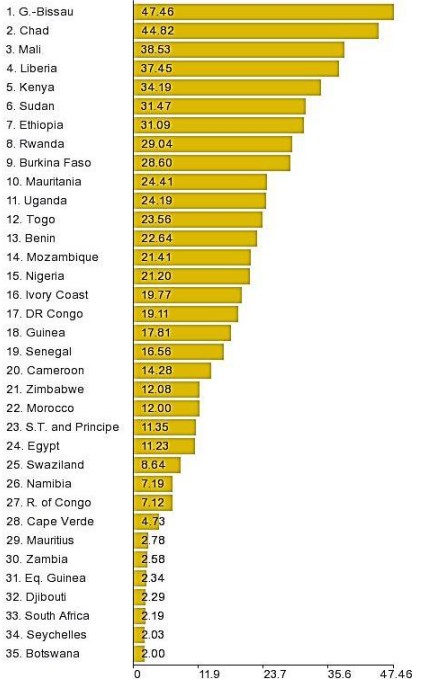

**Figure 1.** Value Added in the Agricultural Sector of African Countries as Percent of GDP. (source: https://www.theglobaleconomy.com/ on data by World Bank, 2018).

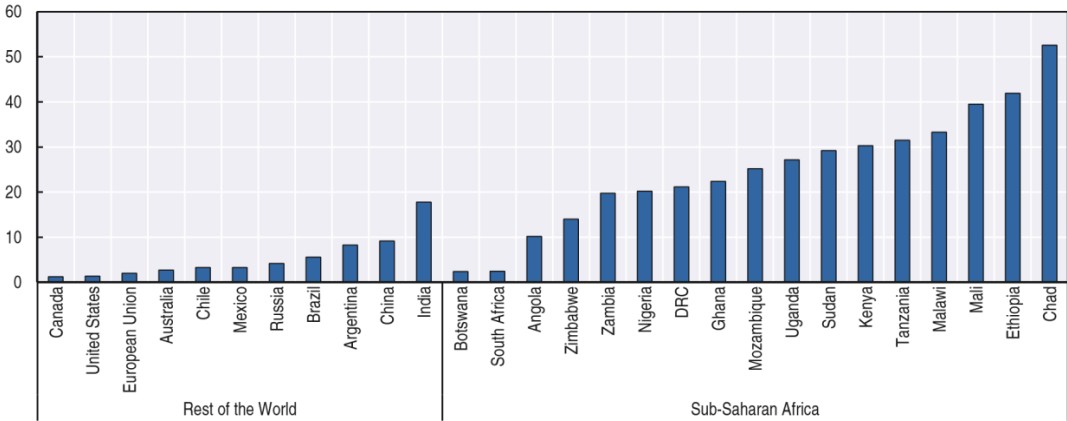

**Figure 2.** Value Added in the Agricultural Sector of the Rest of the World and Sub-Saharan Africa as Percent of GDP. (source: World Bank, 2016).

The transitional and developing countries should be able to develop their economies and simultaneously decrease their GHG emissions, by carrying out and implementing less carbon-intensive technologies to get better farm productivity but also to cut the GHG emissions. Adequate agricultural research systems, new feed processes and technologies are crucial [45]. Consequently, in order to decrease GHG emissions, a correct implementation of resources' transformative approaches (water, fertilizers, fossil fuels, and waste) is necessary especially when considering that countries belong to a global economic system that can determine changes over the world, for example by affecting commodity prices and thus budgetary implications [46].

## 3. Materials and Methods

We use a sample of 29 developing Countries from the FAOSTAT (FAO Statistical Databases United Nations) website [47] and ASTI (Agricultural Science and Technology Indicators) dataset that collects open-access data on agricultural research investment in low/middle-income countries to investigate the effect of R&D (treatment) on GHG emissions (outcome) deriving from dairy cattle farms in the year 2014 (latest accessible data). The choice of the dairy cattle sector depends on the considerable share of emissions of the dairy sector on total emissions from enteric fermentation in sampled countries, although with different situations from country to country (Figure 3).

The choice of analyzed countries was determined by the availability of ASTI datasets related to investments in agriculture. In any case, the selected countries represent a reasonably representative sample of breeding in developing countries both as geographical distribution and as number of animals. In relation to emissions, the methane emission variable ($CH_4$) was chosen to be the greenhouse gas most linked to livestock farming [48]. This emission contributes to 16% of the total greenhouse gas emissions in 2010 [49]. The gases emission depends on the more simultaneous microbial processes of the several cultivations and farming activities [50]. $CH_4$ is released from methane and oxidized by methanotrophic microorganisms, so the variable has been included in relation to these processes [51].

Then, the methane emission variable ($CH_4$), expressed in $CO_2$ equivalent, is our outcome variable (http://www.fao.org/faostat/en/#data/GE/metadata, accessed 10/03/2018). The model uses, as variable treatment, public agricultural R&D investments (ASTI dataset) and, as covariates, livestock of dairy cattle and population. Regarding covariates, livestock of live animals are expressed in number of animals and population data (in millions) refers to the World Population Prospects made by the UN (http://www.fao.org/faostat/en/#data/GE/metadata, accessed 10/12/2018). We closely follow the method of Generalized Propensity Score proposed by Hirano and Imbens [52] for the evaluation of random effects of treatments. Let a random sample of i Country, for $I = 1, \ldots, n$; for each country, i occurs as an array of probable results $Y_i(t)$ for $\Im \in t$, correspondent to the dose-response function for each unit. In the continuous domain, $\Im \in [t_0, t_1]$, while in the binary domain $\Im = \{1,0\}$. The method of Generalized

Propensity Score has the goal of calculating $\mu(t) = E[Y_i(t)]$, which is the Average Dose-Response Function (ADRF).

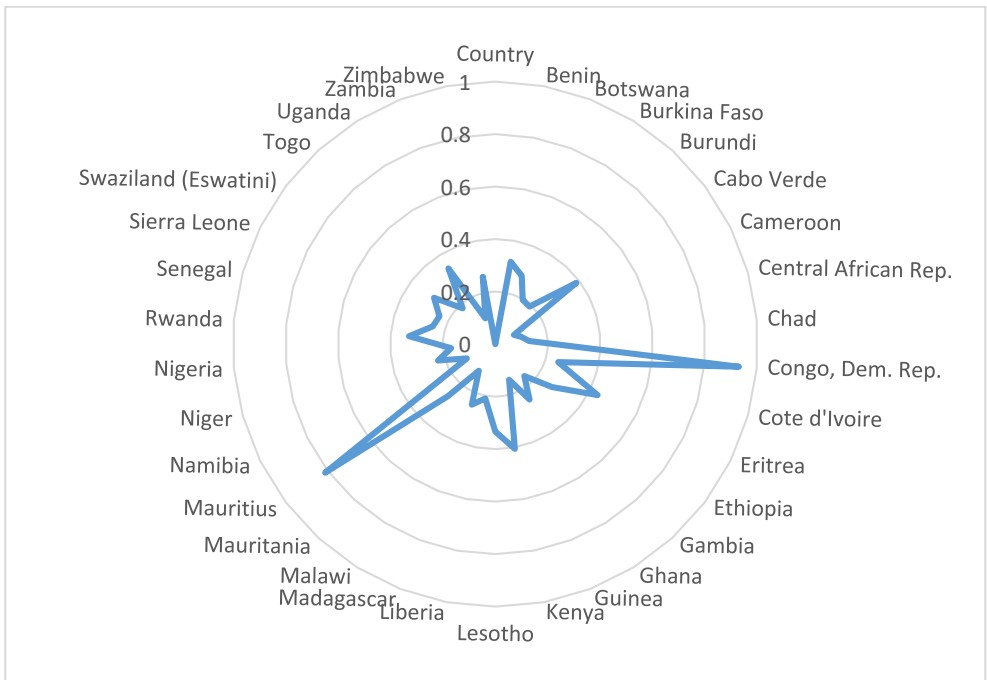

**Figure 3.** Percentage of Dairy Cattle Emissions with Respect to Total Emissions from Enteric Fermentation (source: our elaboration on FAOSTAT data).

Furthermore, the marginal treatment effect function $(E[Y_i(t)] - E[Y_i(t - \Delta t)])/\Delta t$ is estimated. The main advantage of this approach is that it can also detect any nonlinear functional relationships without relying on a priori imposed functional forms since it is a non-parametric estimator [53].

In the context of binary treatments, the work by Rosenbaum and Rubin [54] was based on the unconfoundedness assumption. Hirano and Imbens [52] extended this assumption even in the continuous domain, however they indicate this assumption in terms of weak unconfoundedness. In fact, the joint independence of the probable results is not necessary, while the conditional independence with respect to each level of the treat is needed.

If we indicate with $r(t, X) = f_{T|X}(t|X)$ the conditional density with respect to the corresponding treatment level given the observed covariates, the Generalised Propensity Scores (GPS) is the following casual variable:

$$R = r(T, X) \tag{1}$$

In the case of the propensity score for binary treatments, according to Rosenbaum and Rubin [54], it is important to verify the balancing property. This property implies that within the layers of equal value of $r(t, X)$ at each treatment level, the probability that the received treatment equals this level of treatment, $T = t$, does not depend on the values of covariates.

If this property and the weak unconfoundedness assumption are satisfied, it is possible to use the GPS to remove any bias connected with differences in the observed covariates among groups of units with different levels of treatment. Hirano and Imbens [52] prove if assignment to treatment is weakly unconfounded, given the covariates $X$, it is also weakly unconfounded, given the Generalized Propensity Score. First of all, the conditional expectation of the observed outcome should be evaluated in terms of the treatment function level T and the GPS:

$$\beta(t, r) = E[Y|T = t, R = r] \tag{2}$$

Subsequently, it is necessary to evaluate DRF for level of treatment, by averaging the conditional expectation function according to the GPS by means of the value of treatment: $\mu(t) = E[\beta(t, r(t, X))]$

$$\mu(t) = E[\beta(t, r(t, X))] \tag{3}$$

Let a Gaussian distribution for the treatment by covariates estimated by using OLS (Ordinary Least Squares) methods:

$$T_i | X_i \sim N\left(\beta_0 + \beta'_1 X_i, \sigma^2\right) \tag{4}$$

The GPS has estimated as follows:

$$\hat{R}_i = \frac{1}{\sqrt{2 \prod \sigma^2}} exp\left(-\frac{1}{2\sigma^2}\left(T_i - \hat{\beta}_0 - \hat{\beta}_1 X_i\right)^2\right) \tag{5}$$

Next, we need to make an estimate of the conditional expectation function of outcome by GPS and value of treatment level:

$$E[Y_i | T_i, R_i] = \alpha_0 + \alpha_1 T_i + \alpha_2 T_i^2 + \alpha_3 T^3{}_i + \alpha_4 R_i + \alpha_5 R^2{}_i + \alpha_6 R^3{}_i + \alpha_7 T_i R_i + \alpha_8 T^2{}_i R_i + \alpha_9 T_i R^2{}_i \tag{6}$$

The equation is estimated by means of OLS, by using the observed $T_i$ and estimated GPS. The average outcome at each treatment level of t is as follows:

$$E[Y(t)] = \frac{1}{N} \sum_{i=1}^{N} \left( \begin{array}{c} \hat{\alpha}_0 + \hat{\alpha}_1 t + \hat{\alpha}_2 t^2 + \hat{\alpha}_3 t^3 + \hat{\alpha}_4 \hat{r}(t, X_i) + \hat{\alpha}_5 \hat{r}(t, X_i)^2 + \hat{\alpha}_6 \hat{r}(t, X_i)^3 + \\ \hat{\alpha}_7 t \hat{r}(t, X_i) + \hat{\alpha}_8 t^2 \hat{r}(t, X_i) + \hat{\alpha}_9 t \hat{r}(t, X_i)^2 \end{array} \right) \tag{7}$$

Therefore, the dose-response function is estimated by average potential outcome related to the values of treatment. Furthermore, the non-parametric bootstrap method [52] is used to obtain 95% confidence intervals for estimated dose–response function and for estimated marginal treatment effect.

As in the Propensity Scores Matching, even in the GPS, we need to check the balance of covariate, i.e., whether the estimated expression (5) is appropriate. Hirano and Imbens [52] propose to block the treatment variable and the evaluated GPS. In the generalized propensity scores technique we use a stratification of the treatment variable that produces the grouping of the statistical units in mutually exclusive layers, so that the distribution of the covariates are well identified inside of the same layer [55].

The layers are defined using percentiles and the researcher can decide how many layers should be used in the file analysis, taking into account that the marginal reduction in bias decreases as the number of strata increases. For this purpose, the sample is divided into three ranges according to the public agricultural R&D investments distribution by cutting the 30th and 70th percentile of the distribution and we estimate GPS at the median value of the R&D variable for each of the three sub-samples. For each covariate in subsample we test if the mean is the same of other subsamples. Balancing properties is verified if GPS has been calculated rightly. The estimation of GPS and dose response function were carried out via Stata package 14.0 as in Bia and Mattei [56].

## 4. Results and Discussions

The values of GHG emissions dairy and total cattle, public agricultural R&D investments ASTI expenditures, population, livestock dairy cattle and total cattle, for 29 African countries investigated in 2014 (latest accessible data) are reported in Table 1. Focusing on processed variables, we can notice the sample of the analyzed countries is characterized by a high level of variability, as evidenced by the high values of the respective standard deviations (Table 2).

**Table 1.** GHG Emissions Dairy and Total Cattle, Public Agricultural R&D Investments ASTI Expenditures, Population, Livestock Dairy Cattle and Total Cattle for 29 African Countries Investigated in 2014.

| Country | GHG Emissions ($CO_2$ Equation) Dairy Cattle | GHG Emissions ($CO_2$ Equation) Total Cattle | Public Agricultural R&D Investments ASTI Expenditures (Share of Value Added) ($ USD) | Population (Million) | Livestock (Number Total Cattle) | Livestock (Number Dairy Cattle) |
|---|---|---|---|---|---|---|
| Benin | 516.1338 | 1614.8265 | 0.56 | 10.29 | 2,222,000 | 534,300 |
| Botswana | 318.78 | 1143.3399 | 2.33 | 2.16 | 1,596,605 | 330,000 |
| Burkina Faso | 1255.8 | 6327.5457 | 1.01 | 17.59 | 9,090,700 | 1,300,000 |
| Burundi | 111.09 | 573.9914 | 0.5 | 9.89 | 826,062 | 115,000 |
| Cabo Verde | 6.5727 | 16.9874 | 0.86 | 0.52 | 22,802 | 6804 |
| Central African Rep. | 300.0695 | 2929.6988 | 0.29 | 4.51 | 4,350,000 | 310,631 |
| Chad | 712.908 | 5542.94 | 0.09 | 13.57 | 8,157,404 | 738,000 |
| Congo, Dem. Rep. | 207.048 | 222.222 | 0.43 | 73.72 | 340,000 | 2800 |
| Cote d'Ivoire | 280.14 | 1124.487 | 0.56 | 22.53 | 1,587,000 | 290,000 |
| Eswatini | 130.20 | 444.7762 | 0.74 | 1.29 | 618,000 | 134,788 |
| Ethiopia | 10994.985 | 40501.1804 | 0.24 | 97.37 | 56,706,389 | 11,381,972 |
| Gambia | 54.1424 | 329.6033 | 0.87 | 1.91 | 479,183 | 56,048 |
| Ghana | 290.9891 | 1173.5948 | 0.92 | 26.96 | 1,657,000 | 301,231 |
| Guinea | 603.75 | 4151.049 | 0.29 | 11.81 | 6,074,000 | 625,000 |
| Kenya | 5554.5 | 13690.4584 | 0.78 | 46.02 | 18,247,632 | 5,750,000 |
| Lesotho | 131.376 | 394.4666 | 0.73 | 2.14 | 540,133 | 136,000 |
| Madagascar | 1787.1 | 7222.1688 | 0.13 | 23.59 | 10,198,800 | 1,850,000 |
| Malawi | 106.5034 | 891.9655 | 0.53 | 17.07 | 1,316,799 | 110,252 |
| Mauritania | 352.59 | 1319.325 | 0.45 | 4.06 | 1,850,000 | 365,000 |
| Mauritius | 4.347 | 5.3502 | 4.44 | 1.26 | 6041 | 4500 |
| Namibia | 237.4293 | 1953.9229 | 3.09 | 2.37 | 2,882,489 | 245,786 |
| Nigeria | 2299.08 | 13609.0651 | 0.22 | 176.5 | 19,753,249 | 2,380,000 |
| Rwanda | 275.31 | 834.519 | 0.76 | 11.35 | 1,144,000 | 285,000 |
| Senegal | 610.7651 | 2465.3756 | 1.61 | 14.55 | 3,481,126 | 632,262 |
| Sierra Leone | 115.92 | 488.292 | 0.24 | 7.07 | 692,000 | 120,000 |
| Togo | 56.028 | 303.0109 | 0.17 | 7.22 | 437,390 | 58,000 |
| Uganda | 3381 | 9971.073 | 0.94 | 38.83 | 13,623,000 | 3,500,000 |
| Zambia | 289.8 | 2753.835 | 0.51 | 15.62 | 4,085,000 | 300,000 |
| Zimbabwe | 898.38 | 3462.2504 | 1.4 | 15.41 | 4,868,357 | 930,000 |

Source: our processing of ASTI and FAOSTAT data.

**Table 2.** Descriptive Statistics for Processed Variables.

| Variable | N | Mean | Std. Dev |
|---|---|---|---|
| R&D investments ASTI expenditures (share of value added) ($ USD) | 29 | 0.885 | 0.948 |
| GHG Emissions ($CO_2$ equation) Cattle Dairy | 29 | 737,501.8 | 3,965,463.0 |
| Population (million) | 29 | 23,820 | 36,733 |
| Livestock (number cattle dairy) | 29 | 1,130,806 | 2,328,927 |

Source: our processing of ASTI and FAOSTAT data.

This high variability in the data collected in African countries could be attributed to the impact of global warming on the continent. In fact, according to FAOSTAT (see website 2016), climate change is fostering the process of desertification in some areas of Africa, increasing the differences within the continent and constituting a serious barrier to the development of agriculture, which, however, would need a stable framework useful for planning medium and long-term investments.

Successively, in order to estimate GPS procedure, the first step was the construction of the model of conditional distribution of public agricultural R&D investments (treatment variable) given the covariates (population and livestock), by using OLS. Next, we tested the balancing property of covariates through the two-sided t test.

Table 3 shows that the GPS balancing property is satisfied ($p > 0.05$). Finally, the GPS dose-response function is estimated. The dose response curves are reported in Figure 4a,b, respectively showing for (a) estimated dose–response function, (b) estimated derivative function with 95% confidence bands, as suggested by the methodology of GPS [52].

**Table 3.** Balancing Property for Covariates *.

| | Treatment Interval | | | | | | | | |
|---|---|---|---|---|---|---|---|---|---|
| | [0.09, 0.24] | | | [0.29, 0.56] | | | [0.73, 0.92] | | |
| Covariate | Mean Difference | Std | T Value | Mean Difference | Std | T Value | Mean Difference | Std | T Value |
| Population | 12,724 | 90,755 | 0.140 | −99,062 | 12,457 | −0.795 | 97,987 | 98,338 | −0.996 |
| Livestock dairy | $-1.5 \times 10^6$ | $8.9 \times 10^5$ | −1.64 | $1.8 \times 10^6$ | $9.2 \times 10^5$ | 1.9532 | $2.9 \times 10^5$ | $1.0 \times 10^6$ | 0.289 |

* By two-sided t-test, balancing property is satisfied ($p > 0.05$).

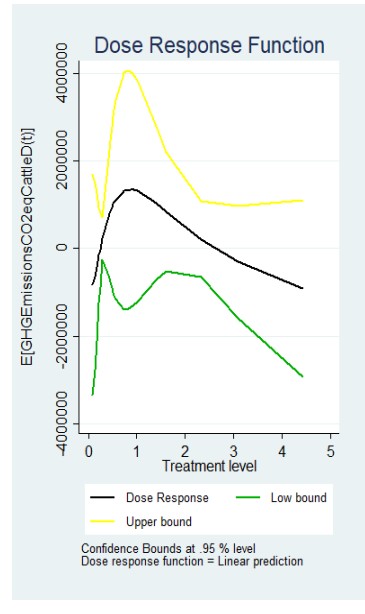

(**a**) dose–response function　　　　　　　　　(**b**) estimated derivative function

**Figure 4.** Estimated (**a**,**b**), and 95% confidence bands. Treatment Variable: R&D Investments. (Source: our processing of FAOSTAT and ASTI data.).

Figure 4a shows a non-linear relationship between public agricultural R&D investments and GHG emissions, highlighted by a curve that after the first quantile of treatment variable shows a decrease. As soon as one country has bigger public agricultural R&D investments expenditure than the other countries, the GHG emissions are lower; in other words, there is a positive response in terms of emissions reductions in 2014 for higher value public agricultural R&D investments. Figure 4b shows estimated elasticity of the average expected response of GHG emissions with respect to public agricultural R&D investments intensity, by means of a constant trend over the first quantile.

Then, at increasing intensities of public agricultural R&D investments, the decrease of emissions appears more relevant. Clearly, other factors come into play at this point, such as the size of the territory, the domestic policies and strategies, the concentrations of companies, the degree of intensive farming of cattle.

## 5. Conclusions and Policy Implications

The agricultural sector in developed countries makes a marginal contribution to the reduction of GHG, while it plays an important role in the environmental sustainability policies of developing countries. For this reason, the paper aimed to estimate the effect of R&D investments expenditure on GHG emissions of cattle dairy in 29 developing countries, by the Generalized Propensity Score method.

As evidenced by the results of the empirical analysis, the investment in research and development certainly has a positive impact on reducing greenhouse gas emissions of the agricultural sector in developing countries. This is in line with previous literature [57–59]. Our results highlight that the amount of emissions decline as soon as public agricultural R&D investments increases. We confirm spillover of knowledge, besides influencing the geographic location choices of enterprises [60], can stimulate the diffusion of eco-innovation [61] and to be induced by greater public investments in R&D.

Data analysis showed how new technology, directly correlated to R&D spending, can play an important role in reducing emissions related to farms in the investigated countries. This result is consistent with the EKC, Environmental Kuznets Curve [4].

Our data analysis has shown how R&D investment results in a positive environmental impact. Policy makers should develop measures in these areas to encourage investment in R&D in agricultural sector by firms.

Policy makers should also consider to improve the overall performance of the breeding industry and to spread new technologies for crop and feed, for example vaccination against rumen methanogens. Policy makers in developing countries could consider these valuable propositions, so that the global GHG emissions depending on the agricultural sector can be allayed by bearing in mind the local context.

Also, in the context of developing countries, factors such as the political, social, demographic features of countries and existing political regimes can systematically determine the extent of expenditure flow. The core policies have to consider the necessity of a transversal firm approach and of the integration of different sustainable practices by elaborating on a suite of successful mitigation practices for specific production systems [62,63].

In order to achieve an absolute reduction in emissions, developed countries have applied several mitigation tools (i.e., energy efficiency, materials use efficiency, materials and products recycling and reuse, industrial symbiosis etc.). The role of agriculture is crucial to feed population and either to decrease its own emissions and to rise carbon removals from the atmosphere in other sectors by means of the replacement of carbon-intensive materials and energy [64]. However, the main challenge for the agricultural sector is to try to increase production in the right way and at the same time reduce environmental impacts [65,66].

In conclusion, farms, which benefit from well-targeted public investment and that therefore will be able to adopt new technologies, will be able to increase their productivity by reducing greenhouse gas emissions. R&D plays a key role in developing countries both in supporting their growth process and in improving environmental sustainability.

Future in-depth studies should broaden these results by assessing in developing countries the impact of public investment in R&D on GHG emissions also in relation to other key factors such as environmental regulation, market demand, environmental management systems and organizational innovations.

**Author Contributions:** Conceptualization: M.F., U.M. and N.F.; Data curation: A.S. and N.F.; Formal analysis: A.S., M.F.; Investigation: M.F. and N.F.; Methodology: A.S.; Resources: N.F.; Software: A.S.; Supervision: A.S. and M.F.

**Funding:** This research received no external funding.

**Conflicts of Interest:** The authors declare no conflicts of interest.

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
