# Peer review of "R&D Expenditure for New Technology in Livestock Farming: Impact on GHG Reduction in Developing Countries"

_sustainability, doi:10.3390/su11247129_

Round 1

Reviewer 1 Report

This research aims at evaluating and understanding if public R&D investments can impact on emissions of business activities in the agricultural sector. To assess this impact, the authors used the so-called Generalized Propensity Scores (GPS), which is a nonparametric counterpart of the Propensity Score Matching.

The analysis suggests that countries show an improvement of their relationship with the environment and the levels of GHG emissions when the public agricultural R&D investments level is bigger.

Overall, the study is interesting but I cannot find the link between R&D investments and digital innovation in livestock farming. More specifically, I understand the connection between R&D investments with GHG emissions but this is not implying a horizontal effect of any type of innovation to GHGs. In other words, this connection does not prove that digital innovations in livestock farming really affect GHG emissions of livestock farming.

However, if the authors can prove this connection, then to my opinion this paper can add to the existing literature. But again, it needs improvement and some issues need to be addressed. Firstly, Figures 1a and 1b are absent. But taking under consideration the text corresponding to these figures, they are important and inextricable part of the results. Therefore, the manuscript should reconsider after these figures will be included in it.  

Some other points that have to be considered by the authors are the following:

The authors should comment/elaborate on the fact that only methane emissions are considered. Methane emissions could be the most important emission but not the only one. What are the cons of omitted other emissions? authors should mention why they choose this specific methodology ADRF/Generalized Propensity Scores (pros and cons) In addition: Line 6. Please clarify why and how you distinguish livestock innovations. Line 38. Incomplete sentence “However, there are few studies from developing world”. Please rewrite this sentence Lines 39-45. This paragraph mention literature that connect economic growth with energy consumption. Please clarify/elaborate the connection of this literature with the purpose of your study. Line 51. Change “consumer” to “consumer’s” Line 62. Please define BRICS Line 63. See comment 3a. Lines 104-105. It would be nice if possible to make to make the representativeness claim more robust by e.g. share of animals or animals’ production Lines 173-175. The authors should explain why they choose three ranges of public investments (and not more ranges) and how they choose these specific cutting points. Line 177. Authors should specify the name of the Stata package they used and corresponding reference. Table 1. What is the Heading: “Tot Pop x1000”? You mean 1000 people? If yes, it may be better to recalculate it as “million people” Lines 192-196. These are not part of the results of the analysis. Please rephrase or move this text to another place of the document

Reviewer 2 Report

1. The paper analyses the impact of R&D expenditure on GHG emission reduction in the livestock farming. Although the topic is very interesting, the authors should make substantial changes to make this paper publishable.

The abstract is very general, that is suggested to be revised.  It is not required to start the paper with highlights. You may incorporate it to either Introduction or Abstract. There is no literature review section in the paper. More information about the role and characteristics agriculture in the analyzed countries would be useful.  The title of the article promises information on new technology and digital innovation, but there is no information on the current state of technology and the capacity and skill level associated with new innovative processes. The information about the contribution of agriculture to greenhouse emission focusing on livestock sector in global and in the countries under analysis is required. Please include more statistics on agricultural GHG emission. This will help the readers get a clear picture about the intensity of the situation before getting into the mathematical approach.  You may add some information such as  - Agriculture is the biggest source of non-co2 anthropogenic emission. Methane emission is around 40%, nitrous oxide emission – 60 % etc.

In Table 1, the headings do not give proper information to the readers. For eg. In the title of table 1, authors have written stocks, if you meant cattle, please do rename it as livestock. It would be nice if you also include the GDP contribution of agriculture for each country. The authors have mentioned dose response curves (fig 1a and 1b),but is not included in the paper. I strongly recommend authors rewrite the conclusion section. Authors have to discuss the implications of their findings in such a way as to highlight the contributions of this paper in this research field. Authors should avoid from the repetitions. In the conclusion, you make a comparison between the GHG emission in developed and developing countries without any supporting evidence for the developed countries. The reducing of CO2 emission in agriculture is a hot topic in developed countries as well, , this question cannot be called marginal.  There are some suggested reading about this topic: Franz Weiss,Adrian Leip Greenhouse gas emissions from the EU livestock sector: A life cycle assessment carried out with the CAPRI model Agriculture, Ecosystems & Environment,  1 March 2012 Ida ML Drejer Storm, EIP -AGRI Focus GroupReducing emissions from cattle farming, STARTING PAPER 14 JANUARY 2016, https://ec.europa.eu/eip/agriculture/sites/agri-eip/files/fg18_starting_paper_2016_en.pdf Lóránt A & Allen B (2019) Net-zero agriculture in 2050: how to get there? Report by the Institute for European Environmental Policy, https://ieep.eu/uploads/articles/attachments/eeac4853-3629-4793-9e7b-2df5c156afd3/IEEP_NZ2050_Agriculture_report_screen.pdf?v=63718575577 https://www.mdpi.com/journal/animals/special_issues/Quantification_and_Mitigation_Strategies_to_Reduce_Greenhouse_Gas_Emissions_from_Livestock_Production_Systems

Kindly include limitations and future research directions.

Round 2

Reviewer 1 Report

In general, authors succesfully replied to my comments.

However, there is still some work to be done:

Line 19: Do you mean: "in its relationship"?

Line 32-33: Is that true? please add a reference

Line 151, 154-155: Title of Figure 1 refers to Figure 2 and vice versa

Figure 3: In some cases countries' captions are on ttop of other countries captions. Please correct

Lines 234-235: equation should be more compound (fit in one line)

Lines 237-238: You use begger font size

Lines 284: Figures need higher resolution and editing. The sentece below legend is not clear. Lower and upper bounds should be differnet fromt the main line (e.g. use different colours or dash lines)

Author Response

Thank you very much for the kindly made revisions. Here are the answers point by point.

Line 19: Do you mean: "in its relationship"?

Yes, exactly, I corrected as suggested

Line 32-33: Is that true? please add a reference

Thanks, I added.

Line 151, 154-155: Title of Figure 1 refers to Figure 2 and vice versa

Thanks, done.

Figure 3: In some cases countries' captions are on ttop of other countries captions. Please correct

Thanks, done.

Lines 234-235: equation should be more compound (fit in one line)

Thanks, done

Lines 237-238: You use begger font size

Thanks, done

Lines 284: Figures need higher resolution and editing. The sentece below legend is not clear. Lower and upper bounds should be differnet fromt the main line (e.g. use different colours or dash lines)

Thanks, done.

Reviewer 2 Report

Thank you for revising this article.
All my suggestions have been taken into consideration.
Beyond my suggestions, the authors have made other significant changes
making the article valuable and also added significant messages
to policy makers.

The article in its present form suitable for publication. Congratulation!

Author Response

Thanks very much for your help!